# Generative Modeling of Individual Behavior At Scale

## Abstract

There has been a growing interest in using AI to model human behavior, particularly in domains where humans interact with this technology. While most existing work models human behavior at an aggregate level, our goal is to model behavior at the individual level. Recent approaches to *behavioral stylometry*—or the task of identifying a person from their actions alone—have shown promise in domains like chess, but these approaches are either not scalable (e.g., fine-tune a model for each person) or not generative, in that they cannot generate actions in the style of each person. We address these limitations by casting behavioral stylometry as a multi-task learning problem—where each *task* represents a distinct *person*—and using parameter-efficient fine-tuning (PEFT) methods to learn an explicit *style vector* for each person. Style vectors are generative: they selectively activate shared "skill" parameters to generate actions in the style of each person. They also induce a latent style space that we can interpret and manipulate algorithmically. In particular, we develop a general technique for *style steering* that identifies a subset of players with a desired style property, and steers a new player towards that property. We apply our approach to two very different games, at unprecedented scale: chess (47,864 players) and Rocket League (2,000 players).

## 1   Introduction

The rapid advances in machine learning in recent years has made it increasingly important to find constructive ways for humans to interact with this technology. Even in domains where AI has achieved proficiency, it is often important to understand how humans approach these tasks. Such an understanding can help identify areas for improvement in humans, develop better AI collaborators or teachers, create more human-like experiences, and more.

A common method for capturing human behavior is behavioral cloning (BC), a form of imitation learning [Schaal, 1996] that applies supervised learning to fixed demonstrations collected for a given task. While traditionally used in domains such as robotics [Florence et al., 2022] and self-driving vehicles [Pomerleau, 1988], BC has seen increasing use in gaming, such as in Counter-Strike [Pearce and Zhu, 2022], Overcooked [Carroll et al., 2019], Minecraft [Schäfer et al., 2023], Bleeding Edge [Jelley et al., 2024], and chess McIlroy-Young et al. [2020].

The above work focuses on modeling human behavior in aggregate, with the goal of developing better AI partners, opponents, and training tools. However, we believe that the most value for such goals can be derived by modeling human behavior at the individual level. To that end, recent results in chess have shown the most promise. McIlroy-Young et al. [2020] used behavior cloning to create a set of models called Maia that match human play at 9 aggregate skill levels. By fine-tuning these models on the data of 400 individual players, they created 400 personalized models that achieve 4-5% higher move-matching accuracy on average [McIlroy-Young et al., 2022]. The authors use these models to perform *behavioral stylometry* with high accuracy, where the goal is to identify which person played a given query set of games; in this case, they simply apply each of the 400 models to the query set

Submitted to 38th Conference on Neural Information Processing Systems (NeurIPS 2024). Do not distribute.

and output the one with the highest accuracy. McIlroy-Young et al. [2021] propose a more scalable approach of training a Transformer-based embedding on the games of each player, and use this to perform accurate stylometry across 2,844 players; in this case, they compute the embedding of the query set of games and match it to the closest player's embedding.

These approaches have different merits. The individual model approach creates a generative model for each player, but it is not scalable and shares only initial (base model) knowledge across the players; adding a new player requires fine-tuning a separate model. The embedding approach is much more scalable: it learns a compact (single-vector) representation of each player in a shared style space, and supports few-shot learning to embed a new player in this space. It cannot be used to generate moves, however, and hence cannot reason about player behavior in practice.

An ideal solution would combine these properties: generative, scalable, shared knowledge, compact representation. Our key insight for achieving this is to view behavioral stylometry as a multi-task learning problem, where each *task* represents an individual *person*. The goal here is to generalize across an initial set of players (tasks) while supporting few-shot learning of new players (tasks). To do this efficiently, we leverage recent advances in parameter-efficient fine-tuning (PEFT) [Ponti et al., 2023, Caccia et al., 2022]. Specifically, we augment an existing BC model with a set of Low Rank Adapters (LoRAs) as well as a routing matrix that specifies a distribution over these adapters for each player. Unlike approaches that train a separate LoRA for each task, this modular design allows players to softly share parameters in a fine-grained manner. We apply this adapter framework to two very different game models (which we create): a modified version of the Maia model for chess, and a Transformer-based BC model for Rocket League, a 3D soccer video game played by cars in a caged arena. (Our models scale beyond the prior art and may be of independent interest.) Our methodology first trains the BC models to convergence across all player data, and then fine-tunes the adapters and routing matrix on per-player data. This encourages the adapters to learn different *latent skills* that explain the variance between players, while each row of the routing matrix induces a weight distribution over these skills. We call each row the *style vector* for the corresponding player.

Style vectors are versatile and powerful. They support few-shot learning which enables stylometry at scale. They induce a generative model for each player that we can run and observe. They induce a shared style space that we can interpret and manipulate algorithmically. Leveraging these properties, we develop a general technique for *style steering* that identifies a subset of players who exhibit a desired style property, and steers a new player towards that property. Our main results include:

1. We perform behavioral stylometry at an unprecedented scale for chess (47,864 players, 94.4% accuracy) and Rocket League (2,000 players, 86.7% accuracy), using a query set of 100 games.
2. Our per-player generative models achieve move-matching accuracy in the range 45-69% for chess and 44-72% for Rocket League, even for players with very few (e.g., 50) games.
3. Style vectors capture a wide diversity of playing styles and strengths. They can be combined, interpolated, and steered, while reflecting consistent changes to play style and strength.

## 2 Background and Framing

We frame behavioral stylometry and per-player generative modeling as a multitask learning problem, to which we apply PEFT methods. In multitask learning [Caruana, 1997, Ruder et al., 2019], we are given a collection of tasks $\mathcal{T} = (\mathcal{T}_1, \ldots, \mathcal{T}_{|\mathcal{T}|})$, each task $\mathcal{T}_i$ associated with a dataset $\mathcal{D}_i = \{(x_1, y_1), \ldots, (x_{n_i}, y_{n_i})\}$. Multitask learning exploits the similarities among related training tasks by transferring knowledge among them; ideally, this builds representations that are easily adaptable to new tasks using potentially few target examples. The premise of this paper is that modeling individual human behavior from a pool of players can be interpreted as a multitask learning problem. In other words, each task $\mathcal{T}_i$ consists of modeling the behavior of a specific player $i$; and dataset $\mathcal{D}_i$ corresponds to the sequence of game actions taken by player $i$. Specifically, an $(x, y)$ tuple denotes a game state $x$ at a specific point in time during game, along with the action $y$ that player $i$ took in this state. For the rest of the paper, we use the notion of *tasks* and *players* interchangeably.

### 2.1 Parameter-efficient fine-tuning

Popularized in NLP, parameter-efficient fine-tuning (PEFT) [Houlsby et al., 2019, Hu et al., 2022, Liu et al., 2022] approaches have emerged as a scalable solution for adapting Large Language Models to several downstream tasks. Indeed, standard finetuning of pretrained LLMs requires updating (and

storing) possibly billions of parameters for each task. PEFT methods instead freeze the pretrained model and inject a small set of trainable task-specific weights, or "adapters".

One such approach is the use of Low Rank Adapters (LoRA) [Hu et al., 2022], which modify linear transformations in the network by adding a learnable low rank shift

$$h = \left( \boldsymbol{W}_0 + \Delta \boldsymbol{W} \right) x = \left( \boldsymbol{W}_0 + \boldsymbol{A} \boldsymbol{B}^T \right) x. \tag{1}$$

Here, $\boldsymbol{W}_0 \in \mathbb{R}^{d \times d}$ are the (frozen) weights of the pre-trained model, and $\boldsymbol{A}, \boldsymbol{B} \in \mathbb{R}^{d \times r}$ the learnable low-rank parameters of rank $r \ll d$. With this approach, practitioners can trade off parameter efficiency with expressivity by increasing the rank $r$ of the transformation.

## 2.2 Polytropon and Multi-Head Adapter Routing

Standard PEFT methods such as LoRA can adapt a pretrained model for a given task. In multitask settings, training a separate set of adapters for each task is suboptimal, as it does not enable any sharing of information, or *transfer*, across similar tasks. On the other hand, using the same set of adapters for all tasks risks *negative interference* [Wang et al., 2021] across dissimilar tasks. Polytropon [Ponti et al., 2019] (`Poly`) addresses this transfer/interference tradeoff by softly sharing parameters across tasks. That is, each `Poly` layer contains 1) an inventory of LoRA adapters

$$\mathcal{M} = \{ \boldsymbol{A}^{(1)} \boldsymbol{B}^{(1)}, \ \dots \ , \ \boldsymbol{A}^{(m)} \boldsymbol{B}^{(m)} \},$$

with $m \ll |\mathcal{T}|$, and 2) a task-routing matrix $\boldsymbol{Z} \in \mathbb{R}^{|\mathcal{T}| \times m}$, where $\boldsymbol{Z}_\tau \in \mathbb{R}^m$ specifies task $\tau$'s distribution over the shared modules. This formulation allows similar tasks to share adapters, while allowing dissimilar tasks to have non-overlapping parameters. The collection of adapters $\mathcal{M}$ can be interpreted as capturing different facets of knowledge, or *latent skills*, of the full multitask distribution.

At each forward pass, `Poly` LoRA adapters for task $\tau$ are constructed as follows:

$$\boldsymbol{A}^\tau = \sum_i \alpha_i \boldsymbol{A}^{(i)}; \ \boldsymbol{B}^\tau = \sum_i \alpha_i \boldsymbol{B}^{(i)} \tag{Poly}$$

where $\alpha_i = \mathtt{softmax}(\boldsymbol{Z} \left[ \tau \right])_i$ denotes the mixing weight of the $i$-th adapter in the inventory, and $\boldsymbol{A}^{(i)}, \boldsymbol{B}^{(i)}, \boldsymbol{A}^\tau, \boldsymbol{B}^\tau \in \mathbb{R}^{d \times r}$. Here, the $\tau$-th row of the routing matrix $\boldsymbol{Z}$ is effectively selecting which adapter modules to include in the linear combination. In our setting, where each task consists of modeling an individual, $\boldsymbol{Z} \left[ \tau \right]$ specifies which latent skills are activated for user $\tau$; we call this their *style vector*. As per Eqn 1, the final output of the linear mapping becomes $h = \left( \boldsymbol{W}_0 + \boldsymbol{A}^\tau (\boldsymbol{B}^\tau)^T \right) x$.

In `Poly`, the module combination step remains *coarse*, as only linear combinations of the existing modules can be generated. Caccia et al. [2022] propose a more fine-grained approach, called Multi-Head Routing (`MHR`), which is what we use in our work. Similar to Multi-Head Attention [Vaswani et al., 2017], the input dimension of $\boldsymbol{A}$ (and output dimensions of $\boldsymbol{B}$) are partitioned into $h$ heads, where a `Poly`-style procedure occurs for each head. The resulting parameters from each head are then concatenated, recovering the full input (and output) dimensions. See A.1 for more details.

**Routing-only fine-tuning.** While LoRA adapters can reduce the parameter cost from billions to millions [Liu et al., 2022], training the adapters for each new task can still be prohibitive when dealing with thousands of tasks. To this end, Caccia et al. [2022] proposed routing-only finetuning, where after an initial phase of pretraining, the adapter modules are fixed, and only the routing parameters $\boldsymbol{Z}$ are learned for a new task. This reduces the parameter cost for each additional task by several orders of magnitude, while maintaining similar performance. We use this method for few-shot learning.

# 3 ML Methodology

In this section, we detail our methodology for creating a generative model of individual behavior that enables our style analyses. Our methodology applies to any behavior cloning scenario with access to human demonstrations from multiple individuals. To demonstrate this generality, we apply it to two very different games: chess and Rocket League. We start with a base model for each and apply the `MHR` adapter framework to it, and then discuss model training and evaluation.

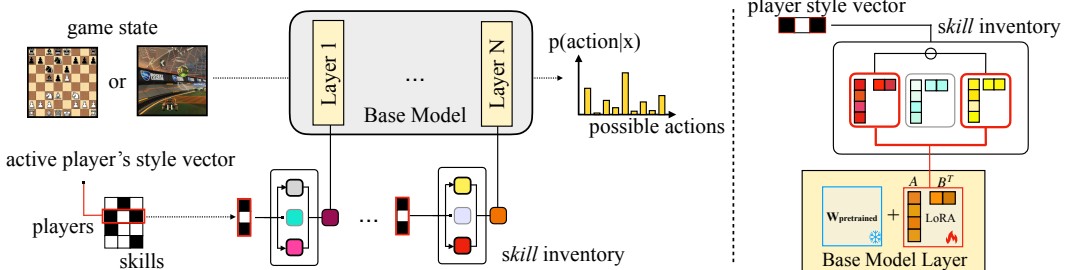

Figure 1: (left) Our overall architecture. We augment a base model with a set of MHR adapters and a routing matrix composed of each player's style vector. (right) Detailed view of an MHR layer, showing a skill inventory of adapters shared across players. The player's style vector specifies which skills are active (in this case, the first and third) to generate the final low-rank weight shift that is applied to the (frozen) base model layer.

## 3.1 Model architecture

For chess, we follow McIlroy-Young et al. [2022] and use the Squeeze-and-Excitation (S&E) Residual Network [Hu et al., 2018] as a base model, but with a deeper and wider configuration (see A.2). At every residual block, an additional 2-layer MLP rescales the residual output along the channel dimension to explicitly model channel interdependencies. The input is a 112-channel $8 \times 8$ image representation of the chess board; the output is the predicted move represented as a 1858-dimensional vector. The total parameters is 15.7M. For Rocket League, we use the GPT-2 architecture from Radford et al. [2019] with a dimensionality of 768, 12 attention heads, and 12 layers. The input is a 49-dimensional vector with game physics information; the output is 8 heads: 5 with 3 bins of [-1, 0, 1] and 3 binary. The model has no embedding layer, as the game data points are passed directly as tokens after processing. The total parameters is 87.7M.

To enable user-based adaptation, we incorporate the `MHR` adapters described in §2.2 into our base models, as illustrated in Fig. 1. In chess, for every linear transformation in the MLP used for channel-wise rescaling, we add an `MHR` layer built of LoRA adapters with rank 16, for a total of $12 \times 2 = 24$ `MHR` layers. We use an adapter inventory of size 32 and a multi-head routing strategy with 8 heads. Therefore, for each user we must learn $32 \times 8 = 256$ routing parameters as their style vector. This yields 5M additional parameters. For Rocket League, we attach the adapters to the fully connected layer of each transformer block, resulting in 12 `MHR` layers of LoRAs with rank 16. We use an inventory size of 16 and 64 heads. This yields 13.8M additional parameters. To facilitate interpretability and style analysis, we use the same routing (style vector) across all `MHR` layers.

## 3.2 Data collection and partitioning

We use data from the largest open-source online chess platform, Lichess.org [Duplessis, 2021], which boasts a database of over 4.8 billion games. We collected Blitz games played between 2013 and 2020 inclusive—these are games with 3 or 5 minutes per side, optionally with a few seconds of time increment per move—and applied the same player filtering criteria as McIlroy-Young et al. [2022]. The resulting dataset comprises 47,864 unique players and over 244 million games. (See A.2 for a discussion on data imbalance.) For Rocket League, we collect data from a large open-source replay database, Ballchasing.com [CantFlyRL, 2024]. We use 2.2 million 1v1 replays from 2015 to mid-2022, totalling several decades of human game play hours at 5 minutes per game. After parsing, each Rocket League game state is a vector holding the player's 3D position, linear and angular velocity, boost remaining, rotation, and team; we also include the opponent's state and the position, linear and angular velocity of the ball. Given a game state, we have to predict the user's throttle, steer (while grounded), pitch, yaw, roll (while aerial), jump, boost, and handbrake. Additional processing was needed to correct for missing aerial controls and inconsistent sampling rates (24-27hz). Our full data processing procedure, including the challenges we faced, are detailed in A.3.

We divide the set of players into a few subsets to support our training methodology. The *base player* set comprises all data and is used to train the base models. The *fine-tuning player* set is used to fine-tune the `MHR` architecture shown in Fig. 1. (For both, we split each player's data into 80/10/10 for train/test/validation.) The *few-shot player* set is used for few-shot learning based on a reference set of

100 games per player. For our chess experiments, to enable a direct comparison with prior work, we create an additional fine-tuning player set consisting of the same 400 players used in those studies. Currently, we treat each player's data holistically, but in principle one could partition a player's data in different ways to perform a finer analysis of their playing style. We explore this in A.4.

### 3.3 Model training and evaluation

**Base model.** We train our base Maia model for chess using data from a base player set of all 47,864 players, treating this as a classification task of predicting human move $y$ made in chess position $x$, given a datapoint $(x, y)$. We use the same loss functions and evaluation criteria as the original Maia work: Maia's policy head uses a cross entropy loss while the value head uses MSE; the output of the policy head is used to evaluate the model's move-matching accuracy.

We train our Rocket League model using a base player set of over 800,000 players, though the vast majority of players have 5 games or fewer. We discretize the actions into 3 bins for throttle, steer, pitch, yaw, and roll, as most of this data is close to 0, -1, or 1. We use binary outputs for jump, boost, and handbrake. A next-move prediction is labelled correct if and only if all of the outputs are correct.

**MHR fine-tuning.** To train the MHR LoRA adapters, we adopt the methodology used in Caccia et al. [2022]: namely, we freeze the base model and fine-tune the MHR layers and routing matrix using data from a fine-tuning player set. Recall that the routing matrix $Z$ has a row (style vector) for each player in the fine-tuning set. Following Ponti et al. [2019], we use a two-speed learning rate, where the style vectors' learning rate is higher than the adapters', to enable better specialization.

For chess, we use two fine-tuning player sets in our experiments, creating two separate MHR-Maia models. The first set comprises all 47,864 players and is used to evaluate behavioral cloning and stylometry at very large scale. The second set is comprised of the same 400 players used by McIlroy-Young et al. [2022], which we use to compare few-shot learning and stylometry results. For Rocket League, we train an MHR-Rocket model on a fine-tuning set of 2,000 players with 100 games each.

**Few-shot learning.** To perform few-shot learning on our MHR models, we perform the "routing-only fine-tuning" described in section 2.2 that additionally freezes all MHR LoRA adapters. Given a few-shot player, we add a (randomly-initialized) new row to $Z$ and fine-tune it on the player's reference set of games, eventually learning a style vector for the player. Using this style vector, we can invoke a generative model of the player and use it to evaluate move-matching accuracy, as described above. To perform stylometry, if the player is a *seen* player (i.e., part of the fine-tuning set), then a matching style vector already exists in $Z$, and we can find it using cosine similarity. Otherwise, if the player is *unseen*, then we simply repeat the few-shot learning process on a query set of games (from the same player), and compare this new style vector to the entries in $Z$.

For chess, (unless stated otherwise), all of our few-shot experiments use the MHR-Maia model fine-tuned on the 400-player set from McIlroy-Young et al. [2022]. For Rocket League, the few-shot player set consists of 1,000 of the 2,000-player set used to fine-tune MHR-Rocket.

**Evaluation.** We evaluate a fine-tuned MHR model in two ways. First, we measure its move-matching accuracy, similar to how we evaluate the base models. However, since our MHR models provide a generative model for each player (invoked through their style vector), we can separately evaluate each player's model by applying it to their test set and measuring move-matching accuracy. The overall move-matching accuracy for the model is simply the average of these per-player accuracies.

Our second evaluation method uses the model to perform behavioral stylometry among all players in the fine-tuning set. This is done by leveraging our few-shot learning methodology (above). That is, given a query set of games from some player, we learn a new style vector in $Z$ for those games via few-shot learning, and compare this vector to every other vector in $Z$. Using cosine similarity as our distance metric, we simply output the player with the highest cosine similarity to the query set vector.

## 4 Style methodology

The style vectors in $Z$ represent distinct distributions over latent skills that give us a starting point for comparing player styles. For example, our stylometry method above uses the cosine similarity of

| Method | \|Query\| | \|Universe\| | \|Query Games\| | Random (%) | Acc. (%) |
|---|---|---|---|---|---|
| *Seen* few-shot players | | | | | |
| McIlroy-Young et al. [2022] | 400 | 400 | 100 | 0.25 | 98.0 |
| McIlroy-Young et al. [2021] | 400 | 400 | 100 | 0.25 | 99.5 |
| MHR-Maia | 400 | 400 | 100 | 0.25 | **99.8** |
| McIlroy-Young et al. [2022] | 400 | 400 | 30 | 0.25 | 94.0 |
| MHR-Maia | 400 | 400 | 30 | 0.25 | **98.8** |
| MHR-Maia | 10000 | 47864 | 100 | 0.002 | **94.4** |
| *Unseen* few-shot players | | | | | |
| McIlroy-Young et al. [2021] | 578 | 2844 | 100 | 0.035 | 79.1 |
| MHR-Maia (100 games) | 10000 | 10000 | 100 | 0.01 | 87.6 |

Table 1: Stylometry accuracy results. *Seen* few-shot players are a subset of the fine-tuning player set, unlike *unseen* players. Numbers for McIlroy-Young et al. [2022] and McIlroy-Young et al. [2021] are borrowed from their respective papers.

these vectors to determine how similar or different players are. However, style vectors also enable much more powerful capabilities, such as the ability to synthesize new (human-like) styles.

To begin, we measure the intra-player consistency of style vectors by splitting a player's dataset into disjoint subsets of varying size, and few-shot learning a style vector for each subset. We then investigate inter-player consistency by merging the datasets of two players and seeing if the style vector trained on the merged dataset is similar to the average of the two player's style vectors.

The latter method actually creates a new playing style that is human-like and yet previously unseen in the world. This suggests a more general approach to style synthesis: interpolate between players using a convex combination of their style vectors. To determine the playing strength of these new players, we can simulate games between them and the players they are derived from. The results of these games can be used to calculate a win rate, which can then be converted to a strength rating.

Currently, our advanced style synthesis techniques focus on chess, where there is a robust mapping between win rates and playing strength (the Elo rating system), and simulating games is cheap. Rocket League simulations are quite costly at present, but in principle the same methodology should apply and we plan to reduce these costs in future work.

In order to make style comparisons more human-understandable, we again exploit the generative nature of our MHR models. Inspired by the concept probing technique used to analyze AlphaZero (a deep RL chess engine) [McGrath et al., 2022], we use a set of human-coded heuristic functions found in Stockfish (a traditional chess engine) to evaluate a player's model. These functions capture concepts such as: king safety, material imbalance, piece mobility, and so on. By invoking a player's model on a fixed set of chess positions and seeing which move they select, we can use this to summarize how much emphasis the player places on the corresponding concepts.

Finally, we combine the above methods to design a simple but general method for *steering* a player's game style towards a specific attribute $a$, such as increasing their king safety, while limiting the changes on other attributes (so as to preserve their style). To achieve this, we first collect a set players $X$ who exhibit high values for attribute $a$—determined, for example, by running their generative models on a fixed set of game states. We then extract the common direction among these players, by averaging their style vectors and subtracting the population average. This yields a *style delta vector* that can be added to any player's style vector to elicit the desired change.

## 5   Experiments

In this section, we demonstrate two main findings. First, MHR-Maia performs competitively with prior methods for behavior cloning and stylometry in chess, while achieving unprecedented scale. We also show that our approach can be applied to Rocket League, for both stylometry and move prediction. Second, we show that explicitly capturing style vectors allows us to reason about and perform arithmetic operations on generated behaviors.

## 5.1 Behavioral Stylometry

In this section, we show that our models perform competitively with previous behavioral stylometry methods for both seen and unseen players. Here, the goal is to predict the player who produced a given set of games. We compare to individual model fine-tuning [McIlroy-Young et al., 2022], fitting a pre-trained Maia to the data from a single player, and to a Transformer-based method [McIlroy-Young et al., 2021], which embeds players in a 512-dimensional style space based on their gameplay. All reported accuracies are top-1 unless stated otherwise.

To perform stylometry on a query set of games, McIlroy-Young et al. [2022] suggest measuring the move-matching accuracy of each available fine-tuned model and selecting the best performing model. As seen in Table 1, this procedure works well, but is tremendously expensive—requiring computationally intensive inference calls on the entire query set for every candidate player.

In contrast, both the Transformer-based method and MHR-Maia scale much better to large numbers of players. The Transformer-based method needs only to condition on these games to produce a vector, while MHR-Maia needs only to fit a new vector. In either case, the produced vectors need only be matched to those in the player set, e.g., using cosine similarity. Table 1 compares both approaches, showing that MHR-Maia performs competitively or better, on a much larger universe. To do this, we use few-shot learning to compute style vectors for 10,000 players based on their 100 game reference sets, then fit new style vectors for each player based on their respective query sets. Note that the individual model fine-tuning method is omitted from the larger few-shot study due to scalability reasons. The Transformer-based method can scale, but it is not a generative model.

For Rocket League, to the best of our knowledge, we are the first to attempt stylometry. We report player identification results averaged over the few-shot player set. For each prediction, our MHR-Rocket approach must correctly identify each of the 1,000 players among a pool of 2,000 players. Yet, it reaches an accuracy of **86.7%** (random: 0.05%), showcasing the validity of our approach.

## 5.2 Move generation

Here we compare the efficacy of our method to using individually fine-tuned models for each player. Fine-tuning individual models generally results in superior results compared to PEFT methods, as the increased parameter count produces more expressive models. However, they are also more computationally intensive to train and store. That said, in the domain of modeling individual behavior in chess, MHR-Maia is able to perform comparatively well despite using a much smaller parameter budget. Figure 2 shows that MHR-Maia matches individual model fine-tuning over a wide range of game counts. The base model is frozen for all game counts in MHR-Maia. The model has already learned the set of skills required to differentiate the players, all that is

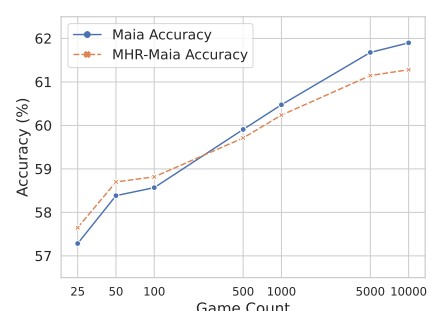

Figure 2: Accuracy at various game counts of the individual models (Maia) and our method (MHR-Maia).

needed with very few-shot learning is to find a proper recombination of the learned skills within the new style vectors. The Transformer-based method is omitted, as it is incapable of generating moves.

For Rocket League, we compare the next move prediction of our base model, with MHR-Rocket, to validate that our user-based conditioning generates better predictions. We find that, on average, MHR-Rocket increases the next move prediction from **53.1%** to **56.1%**.

## 5.3 Analysis of style vectors

In this section, we explore the consistency of our style vectors across different players and datasets.

**Consistency across a single player.** To showcase intra-player consistency, we first partition 50 players' datasets into disjoint subsets. We use 50 splits for chess and 20 for Rocket League. The subsets are sampled across a wide range of dates, opposing players, and playing sessions. Next, we train a style vector for every split across all players. We find that vectors corresponding to the same player will be similar to each other, and have low similarity with the other players and general population. This is visualized in Figure 3. This suggests that our neural network is able to find

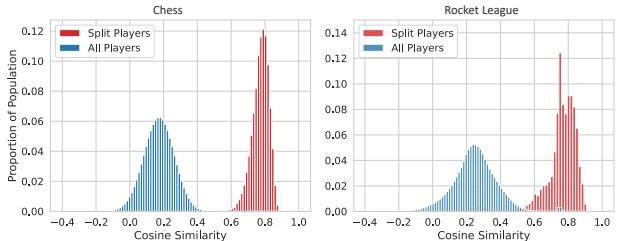
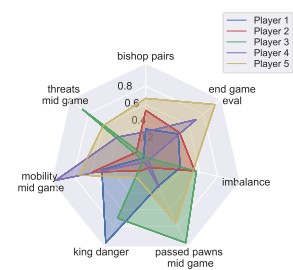

Figure 3: Cosine similarity between style vectors learned from different partitions of the same player (red) vs across different players (blue).

Figure 4: Comparing player styles using human-interpretable evaluation metrics.

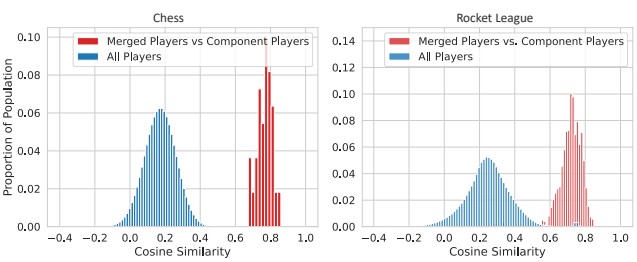
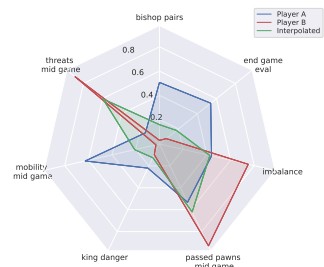

Figure 5: Cosine similarity between averaged style vectors of two players, and the learned style vectors on their merged datasets (red) vs across the full population (blue). The style of an intermediate player (green) is shown along with the two component players (blue and red) on the right.

distinct tendencies for each player. To confirm, we sampled 5 random chess players, predicted their preferred move across $2^{17}$ positions, and measured a series of Stockfish evaluation metrics per player. Figure 4 shows the distribution of these metrics for each player, demonstrating that these vectors store a wide diversity of styles.

**Consistency across merged players.** To parse out whether we can generate new styles using this information, we merged two players' datasets together to generate a new set with the tendencies of both players, measuring inter-player consistency. We then compared this new set of vectors to a different set of vectors generated by simply averaging the style vectors of the player pair. As seen in Figure 5 (left and center), vectors with the same two source players have very high similarity in both chess and Rocket League. We then sampled a random pair in the merged dataset, created a new player by averaging the two players' vectors, and recorded their gameplay according to the previous section. The results are visualized in Figure 5 (right), showing that the new player (green) has an intermediate playing style to the source players (red, blue).

## 5.4   Synthesis of new styles

**Convex combinations.** We show that interpolating between skill vectors results in a player whose level is a weighted average of the interpolated players. Here, we take 100 pairs of learned player vectors, such that one item in the pair corresponds to a strong player and the other to a weaker player. We then gradually interpolate between the weak and strong player as $(1 - \lambda)u_w + \lambda u_s$, $0 \leq \lambda \leq 1$, where $u_w$ and $u_s$ are respectively vectors representing the weak and strong player. For each value of $\lambda$ we simulate 1,000 games between the interpolated vector and $u_s$, the stronger player.

Figure 6 plots the win rate of the interpolated player as a function of $\lambda$ for each player pair we considered. This plot demonstrates that win rate progresses in a roughly linear fashion, starting off winning infrequently against the stronger player and eventually winning roughly half the time as the interpolated player converges to the stronger player.

**Directly steering player style.** Finally, we directly control the playing style of a player by creating skill vectors according to the procedure described in 4. We choose players in our chess dataset with high (>2 std) bishop pair utilization, and separately players with high king danger. Figure 7 shows

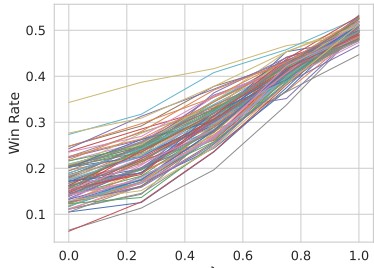

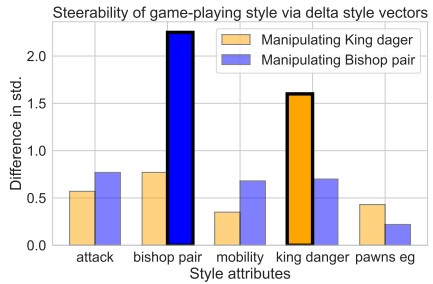

Figure 6: Winrate as a weaker player is interpolated with a stronger player as a function of $\lambda$.

Figure 7: Modifying the a player's style towards a specific attribute

the change in 2,000 randomly sampled player's stockfish evaluations after adding the skill vector corresponding to each heuristic to their style vectors. Indeed, we see that the player's style is steered towards the attribute in question, with model impact on other attributes.

# 6  Related Work

**Stylometry and player style modeling.**  Originally referring to performing author attribution via statistical analysis of text [Tweedie et al., 1996, Neal et al., 2017], stylometry has since come to refer to the general task of identifying individuals given a set of samples or actions, and has found broad application for tasks such as handwriting recognition [Bromley et al., 1993], speaker verification [Wan et al., 2018], identifying programmers from code [Caliskan-Islam et al., 2015], determining user age and gender from blog posts [Goswami et al., 2009], and identifying characteristics of authors of scientific articles [Bergsma et al., 2012]. In the context of gaming (covered in the introduction), stylometry is closely related to playstyle modeling, where the goal is to associate a player with a reference style, such as by building agents representative of different playstyles and find the closest behavioral match [Holmgård et al., 2014], or gathering gameplay data and applying methods such as clustering [Ingram et al., 2022], LDA [Gow et al., 2012], Bayesian approaches [Normoyle and Jensen, 2015] and sequential models [Valls-Vargas et al., 2015] to identify groups of players with similar styles. Unlike our work, these approaches focus on aggregate playstyles, and do not learn generative models that can be conditioned on an individual's style.

Our method for style synthesis is inspired by earlier work on vector arithmetic with embeddings [Church, 2017], as well as recent work on steering multiask models with task vectors [Ilharco et al., 2023]. Finally, our steering method is reminiscent of Radford et al. [2016], which manipulates the model's latent space to generate images containing specific attributes.

**Parameter-efficient adaptation**  Approaches for efficient adaption of a pretrained model can be broadly grouped in two categories. Adapter based methods inject new parameters within a pretrained model, and only updates the newly inserted parameters while keeping the backbone fixed. Houlsby et al. [2019] defines an adapter as a two-layer feed-forward neural network with a bottleneck representation, and are inserted before the multi-head attention layer in Transformers. Similar approaches have been used for cross-lingual transfer [Pfeiffer et al., 2020]. Adapters have also been used in vision based multitask settings [Rebuffi et al., 2017]. More recently, Ansell et al. [2022] propose to learn sparse masks, and show that these marks are composable, enabling zero-shot transfer. Lastly, Hu et al. [2022] learn low-rank shifts on the original weights, and [Liu et al., 2022] learns an elementwise multiplier of the pretrained model's activations. Adapters have also been used in multitask settings. Chronopoulou et al. [2023] independently trains adapters for each task. In order to transfer to new tasks, the authors merge the parameters of the adapters of relevant training tasks.

# 7  Conclusion

We show that individual player behavior can be modeled at very large scale in games as different as chess and Rocket League. We cast this problem in the framework of multi-task learning and employ modular PEFT methods to learn a shared set of skills across players, modulated by a distinct style vector for each player. We use these style vectors to perform behavioral stylometry, analyze player styles, and synthesize and steer new styles.

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

# A  Appendix

## A.1  Multi-Head Adapter Routing

In `Poly`, the module combination step remains *coarse*, as only linear combinations of the existing modules can be generated. Caccia et al. [2022] propose a more fine-grained module combination approach, referred to as Multi-Head Routing (`MHR`). Similar to Multi-Head Attention [Vaswani et al., 2017], the input dimension of $A$ (and output dimensions of $B$) are partitioned into $h$ heads, where a `Poly`-style procedure occurs for each head. The resulting parameters from each head are then concatenated, recovering the full input (and output) dimensions. This makes the module combination step *piecewise linear*, with a separate task-routing matrix $Z$ learned for each head.

Formally, a `MHR` layer learns a 3-dimensional task-routing tensor $\mathbf{Z} \in \mathbb{R}^{|\mathcal{T}| \times |\mathcal{M}| \times h}$. The 2D slice $\mathbf{Z}_{:,:,k} \in \mathbb{R}^{|\mathcal{T}| \times |\mathcal{M}|}$ of the tensor $\mathbf{Z}$ denotes the distribution over modules for the $k$-th head, and $W[k] \in \mathbb{R}^{\frac{d}{h} \times r}$ the $k$-th partition along the rows of the matrix $W \in \mathbb{R}^{d \times r}$. The adapter parameters $A^\tau \in \mathbb{R}^{d \times r}$ for task $\tau$, and for each adapter layer, are computed as (similarly for $B^\tau$):

$$A_k^\tau = \sum_j \alpha_{i,k} \cdot A_j[k] \ \text{ with } \ A_k^\tau \in \mathbb{R}^{\frac{d}{h} \times r}, \tag{MHR}$$

$$A^\tau = \texttt{concat}(A_1^\tau, \dots, A_h^\tau),$$

where $\alpha_{i,k} = \texttt{softmax}(Z[\tau,:,k])_i$. Importantly, the number of LoRA adapter parameters does not increase with the number of heads. Only the task-routing parameters linearly increase with $h$ for `MHR` vs. `Poly`. However, this cost is negligible as the parameter count of the routing matrices is much smaller than for the LoRA modules themselves.

## A.2  Maia Architecture/Data

Our base Maia architecture follows McIlroy-Young et al. [2022] and uses the Squeeze-and-Excitation (S&E) Residual Network of [Hu et al., 2018]. At every residual block, channel information is aggregated across spatial dimensions via a global pooling operation. The resulting vector is then processed by a 2-layer MLP, with a bottleneck representation compressing the number of channels by $r$. The output of this MLP is a one-dimensional vector used to scale the output of the residual block along the channel dimension. We use 12 residual blocks containing 256 filters, and a bottleneck compression factor of $r = 8$. We note that this differs from the base Maia model in McIlroy-Young et al. [2022], which uses 64 filters and 6 residual blocks.

While our dataset has a median game count of 3,479 games, many players may have as few as 10-50 games, implying some degree of data imbalance. Our evaluation of few-shot learning shows that 100 games is sufficient to learn the style vector of an unseen player. However, one might still ask how accurately such a style vector is given a very small number of games. To explore this, we first split a player into disjoint sets of 10, 25, 50, 100, 500, and 1,000 games. We then train a style vector on each set. As a baseline, we train a style vector on 10,000 games and track the cosine similarity of the smaller-set style vectors relative to this baseline vector. We show the results in Figure 8.

## A.3  Rocket League Architecture/Data

The 1v1 replays dataset was scraped over the course of several weeks from the Ballchasing.com API using the Grand Champion subscription tier, though the API does have a slower free tier. This API yields raw game replays, which are uploaded by users either manually or using a community-made plugin for the game. The replays are in a binary format which must be parsed using community-made projects such as Carball [SaltieRL, 2024].

The Carball library allows us to convert the binary replay format to a more standard CSV format, which we save to a Cloud binary blob storage. The data present in both is a lossy reconstruction of game states, and requires some processing to be usable. In particular, the data is sampled at an inconsistent rate (varying between 24hz and 27hz), contains repeated physics ticks, and is missing action data for aerial controls (pitch, yaw, roll).

We resolve the issue of sampling rate and repeated ticks by removing repeated ticks, and doing a time-weighted resampling and interpolation to a standard 10hz for model training, though we found

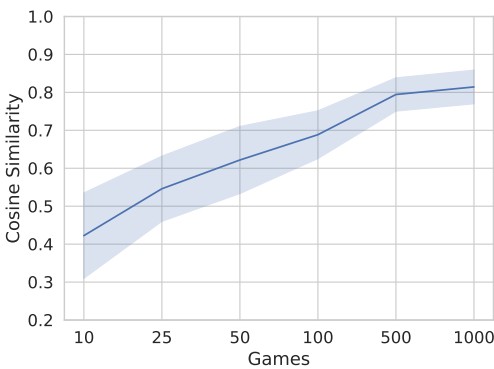

Figure 8: Cosine similarity of style vectors trained with varying game sizes compared to a style vector trained with 10,000 games, run on 50 players.

that 30hz also works well. Note that the actual game physics ticks occur at 120hz, so any value aligned with this should work. Without these changes, the model performs extremely poorly and is unable to navigate the arena.

We resolve the issue of missing aerial controls through the physics-based solver present in the Carball library. The estimation of these controls is not perfect, but it is sufficient for our purposes. Some previous community work has used inverse dynamics [Braaten, 2022] trained from rollouts of in-game bots to solve for these actions, though we opted to not use this due to the inconsistency in replay data sampling.

The data returned by the CSVs are fairly large, messy, and inconsistent. We apply the following transformations to the dataframe to bring the values closer to 0:

- Divide position by 2300
- Divide linear velocity by 23000
- Divide angular velocity by 5500
- Divide boost by 255
- Encode rotation Euler angles according to Zhou et al. [2020]

Additionally, when turning the data into tokens for use in our model, we add in an extra dimension to represent the team, and concatenate the opponent's data points along with the position, linear and angular velocity of the ball. We complete all of these transformations at runtime.

We also have to align the data returned by the simulators for Rocket League with the data used to train the model, RLBot [RLBot, 2017] and RLGym [Emery, 2021]. Along with including an extra dimension to represent the team, we apply the following transformations to all samples obtained from the game:

- Divide position by 2300
- Divide linear velocity by 2300
- Divide angular velocity by 5.5
- Divide boost by 100

The skill distribution of the players in our dataset can be found in Figure 9.

## A.4 Implicit Stationarity Assumptions

Most of the existing work in chess assumes that a player remains stationary over time and across gameplay situations. However, in reality, a player's style may depend on the type of opponent they are facing, which opening is used, which stage of the game they are in (opening, middle, endgame), and so on. For instance, McIlroy-Young et al. [2021] observe that stylometry accuracy drops when

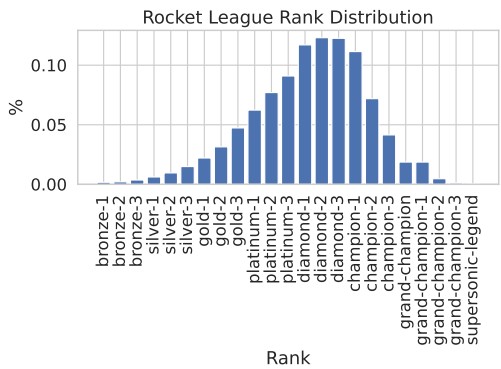

Figure 9: Skill distribution of Rocket League players in our dataset.

removing the opening (e.g., the first 15 moves) moves, suggesting that the opening has an outsized effect on style identification. Our approach does not rely on these assumptions and can in principle be applied to arbitrary subsets of a player's data. For instance, one could split a player's data into opening, middlegame, and endgame moves and train a separate style vector for each. One could further split the data based on which defense the opponent uses, what time of the day it is, etc.. Despite treating players holistically and avoiding any splits of their data, we are still able to capture the peculiarities of each individual's playing style and perform stylometry with high accuracy. This also enables us to compare our results to those of prior work, which also treats player data holistically.

## A.5 Delta Style Vector Computation

---

**Algorithm 1** Style Delta Vector computation

---

**Input:**
$X$ : Style vectors of top-k players for attrib. $a$;
$P$ : Style vectors of all players in population
**Output** $\Delta_a$: Style delta vector for attr. $a$

$V_a = \text{mean}(X, \texttt{axis} = \text{'players'})$
$V_P = \text{mean}(P, \texttt{axis} = \text{'players'})$
$\Delta_a = V_a - V_P$
**Returns** $\Delta_a$

---

