# OpenReview forum: "Generative Modeling of Individual Behavior at Scale"
_NeurIPS.cc/2024/Conference — Submitted to NeurIPS 2024_

### Official Review · Reviewer_tLZq · 2024-06-16

**Soundness:** 3
**Presentation:** 3
**Contribution:** 2
**Rating:** 4
**Confidence:** 3

**Summary:**

This paper adapts Maia, a group-level model and variant of the AlphaZero model proposed by McIlroy-Young et al. (2020), to function at an individual level. The authors focused on scaling the fit to individual behavior using techniques like LoRA for fine-tuning LLMs. Moreover, the learned embeddings were shown to encode decision-making styles that correspond to individual behaviors. Linear combinations of these embeddings can result in desirable decision-making styles, creating synthetic styles.

**Strengths:**

This is an interesting paper that treats fitting individual data as a task in a multi-task learning framework. Recent fine-tuning techniques for LLMs have shown promise in adapting a base model, which captures group-level behavior, to individual-level data on a large scale, accommodating hundreds of thousands of individual datasets.

**Weaknesses:**

The paper lacks a proper benchmark for comparing the accuracy gains from their proposed method. Hypothetically, if the datasets they considered (Chess and Rocket League) exhibit no individual variability in behaviors, meaning all individuals behave identically to the group-level agent, we should expect no accuracy gain whatsoever from adapting a group-level base model to individual-level data. Conversely, if significant variability stems from individual differences, we should expect a substantial gain. Currently, the 0.4% and 4.8% accuracy gains reported in Table 1 are not compared to any benchmark that considers the maximal possible variability that could be extracted from individual-level data. This leaves readers uncertain about the actual gains achieved by the method. It would be beneficial to establish a theoretically optimal benchmark for maximal possible accuracy gains across all datasets.

**Questions:**

Is it possible to use steering or synthetically created new styles to predict human decisions? This approach might provide a stronger test for the learned style vectors that capture human-like behaviors.

**Limitations:**

This paper did not explicitly include a section for Limitation and Future Research.

---

> ### Author Rebuttal · Authors · 2024-08-07
>
> Thank you for your insightful comments and questions! We respond to each one below, and in some cases refer to the general responses above.
>
> **Benchmarking and comparing accuracy gains**
>
> Fig. 2 compares MHR-Maia (our approach) to individually fine-tuning a model for each player.  We believe the latter provides a reasonable benchmark for the accuracy gains that are possible from training on each player’s data. In terms of what these gains are relative to a group-level base model, both our work and prior work (e.g., Fig. 4 in [F]) show that individual model fine-tuning can increase move prediction accuracy by 4-5% on average, implying substantial variability in individual behaviors. For Rocket League, we did not train an individual model for each player, but we report in Section 5.2 that MHR-Rocket increases move prediction accuracy by 3% on average. Our style analysis for chess in Section 5.3 connects style vectors to human-interpretable properties like king safety, material imbalance, etc.; Figs. 4 shows that randomly selected players exhibit substantial variability in these properties.
>
> We note that although our work and prior work has shown that players exhibit substantial individual differences, fitting models that exhibit these differences becomes increasingly difficult as we increase the number of players and decrease the number of model parameters. Our MHR models push in both directions, scaling to (tens of) thousands of players while using only a fraction of additional model parameters relative to the base model.
>
> **Using synthetically created styles to predict human decisions**
>
> For players that we train our MHR models on, or for new players we do few-shot learning on, the style vectors we obtain can be used to predict the decisions of the corresponding individual. For style vectors that have been steered or synthesized from existing players, the resulting vectors represent new styles that may not map to an existing human player. In essence, the purpose of steering is to change the player. One could ask the question of how to steer a player in a *human-like* way – e.g., by observing how a player has changed over time using the human-interpretable metrics from our style methodology (Section 4), and then steering their original style vector along these properties. We believe this is an interesting direction to explore in future work.
>
>
> [F] McIlroy-Young et al. Learning Models of Individual Behavior in Chess. KDD, 2022

---

> > ### Comment · Reviewer_tLZq · 2024-08-12
> >
> > Thank you for the response. Most of my concerns have been adequately addressed, and I am inclined to increase my score to 5.

---

> > > ### Author Response · Authors · 2024-08-12
> > > **Response to reviewer tLZq**
> > >
> > > Thank you for your response! If you feel we have adequately addressed your concerns and are inclined to increase your score to a 5, we would very much appreciate it. Please let us know if anything else is needing of clarification.

---

### Official Review · Reviewer_ALah · 2024-06-24

**Soundness:** 3
**Presentation:** 4
**Contribution:** 3
**Rating:** 7
**Confidence:** 4

**Summary:**

The paper explores modelling user behaviour for chess and rocket league using a PEFT-based method, wherein users are modelled using a composition of MHR adapters. The authors evaluate their approach both for predicting which player played a particular game and for predicting the next move of a given player, and find that their MHR-based approach outperforms prior work in both cases. Further analysis shows that the MHR-based approach further allows steering and combining user vectors to produce new styles, and that the vectors are able to capture a wide diversity of strategies.

**Strengths:**

The approach is simple but intuitive, applying MHR-style adapters to stylometry and exploring the benefits of this approach. The approach does seem to result in improved performance over prior work, and the analysis of the adapters showing that they can be combined to create new styles, or ‘steer’ a user adapter towards a particular style, is interesting, and highlights that using MHR for learning style vectors allows for interesting analyses/applications. The experimental setup seems reasonable, and the authors show their approach works across more than just chess (although chess is the main focus of the work).

**Weaknesses:**

- The MHR approach does seem to underperform full-finetuning when there is a large number of games available, as shown in Figure 2, and as expected for a PEFT-based method.
- While older stylometry work [1] was able to produce user vectors by just performing inference (albeit not being generative), this approach requires training, and so might be more computationally expensive as a result (PEFT training is cheap, but still requires computing the full backward pass to compute gradients for the adapters). Grounding some of the discussion in section 5.1 with compute estimates (e.g. estimated FLOPs) would be useful.
- The novelty is somewhat limited, as the primary method (MHR adapters, linearly interpolating between adapters) has been explored in prior work. However, I think that the application of this idea to a new domain (user identification/move generation) is still novel and interesting.

[1] R. McIlroy-Young, Y. Wang, S. Sen, J. Kleinberg, and A. Anderson. Detecting individual decision making style: Exploring behavioral stylometry in chess. Advances in Neural Information Process439 ing Systems, 34:24482–24497, 2021.

**Questions:**

1. I’m curious about the compute cost of your approach against the other methods. Taking into account training and inference, what are the differing FLOPs costs? Using adapters is typically useful for reducing memory costs (since you don’t need to store optimizer states for all parameters), but still requires a similar number of FLOPs to train.
2. In the Polytropon paper, they attempt to analyse how tasks are distributed among the skill vectors. While this might be a bit trickier with MHR, I’m curious if you observe that if the skill vectors learnt through MHR training correspond to any human-interpretable features? This would be an interesting analysis!

**Limitations:**

The authors address the limitations of their approach reasonably.

---

> ### Author Rebuttal · Authors · 2024-08-07
>
> Thank you for your insightful comments and questions! We respond to each one below, and in some cases refer to the general responses above.
>
> **Comparison to individual model fine-tuning**
>
> We agree that for larger numbers of games, MHR-Maia is expected to have lower accuracy than individual model fine-tuning. However, we note that MHR-Maia has very competitive accuracy – within 1 percent of individual model fine-tuning – even for players with 10,000 games (a tiny fraction of the total population). Furthermore, MHR-Maia does this at a fraction of the cost, as discussed below. We also note that the baseline we are comparing against is our own version of Maia, which achieves higher accuracy than the current state-of-the-art, and hence is a more competitive baseline.
>
> **Cost comparisons**
>
> We do not have exact FLOP numbers, but we do have estimates of the costs of training and inference across different methods. On a machine with 24 CPU cores, 216GB of memory and 1 A100 80GB GPU, the individually fine-tuned models tested in Fig. 2 require an average of **around 20 A100-minutes per player** to train. MHR-Maia required roughly 7 A100-days for 47,864 players, or an estimated cost of **around 12.63 A100-seconds per player**. The inference costs were roughly equal, with MHR-Maia being marginally more expensive due to the added parameters. The improvement in training cost is due to lower memory usage allowing for larger batch sizes, fewer computations in the backward pass, and being able to share learned skills within the LoRAs (which individual model fine-tuning cannot do).  Additionally, our few-shot experiments for stylometry require only enough training time to learn a re-weighting of the shared LoRA skills (in the style vectors), which is orders of magnitude faster than full-model fine-tuning,  and we can use very high learning rates (up to 1e-2) for the style vectors to further speed up training time.
>
> **Comparison to related work**
>
> Please see our discussion of costs above. Note that the method in [D] still requires training on millions of player games to learn an embedding space, which can then be used for inference – i.e., mapping players games to vectors in this embedding space. Although their method was only applied to 2,844 players, we believe it has similar scalability to our approach, while lacking the benefits of a generative model.
>
> **Novelty and new capabilities**
>
> Please see our general response above.
>
> **Connecting skill vectors to human-interpretable features**
>
> We agree that this kind of analysis is indeed interesting! Our analysis of style vectors in Section 5.3 gets at this question, by connecting style vectors to human-interpretable properties in chess like king safety, material imbalance, bishop pair, etc. (the Stockfish metrics). This allows us to compare player styles along human-interpretable properties, as shown in Figs. 4 and 5. Our style steering method goes a step further and identifies which style vector components correspond to a particular human-interpretable property – what we call a “style delta vector”. A style delta vector can be added to an existing player’s vector to steer their style towards that property.
>
> Beyond this, new work is needed to explicitly map style vector dimensions to human-interpretable features. A major challenge in doing so is that, based on experiments we have run, style vector dimensions representing overlapping human features, and the human features themselves are not easily separable. For example, we used our mapping of style vectors to human-interpretable features to run a LASSO regression aimed at identifying which dimensions contribute most to each feature, inspired by the concept probing technique used to analyze AlphaZero [E]. We found that many style vector dimensions contribute to each feature, with enough overlap across features to make it difficult to separate the dimensions in a meaningful way. Another approach is to incorporate human-interpretable features during the MHR training process itself (imposing structure on the style vector dimensions), but this is beyond the scope of the current paper.
>
> [D] McIlroy-Young et al. Detecting individual decision-making style: Exploring behavioral stylometry in chess. NeurIPS, 2021.
>
> [E] McGrath et al. Acquisition of chess knowledge in AlphaZero. PNAS, 2022.

---

> > ### Comment · Reviewer_ALah · 2024-08-11
> > **re: rebuttal**
> >
> > Thank you for the detailed rebuttal! I'm keeping my positive score as-is, and my concerns have mostly been addressed.

---

### Official Review · Reviewer_JjNr · 2024-07-12

**Soundness:** 4
**Presentation:** 4
**Contribution:** 3
**Rating:** 6
**Confidence:** 5

**Summary:**

Authors found a way to obtain player stylometry using BC on a massive amounts of data. Their idea uses multiple LoRAs and routing matrix to obtain a style vector for each player.

**Strengths:**

- Very interesting technical solution and innovative way to use LoRA and routing matrix to obtain player style vectrors using behavioural cloning.
- I think there is a large potential in using this technique in analyzing large game play databases. Possibly to glean new insights.
- Potential also exists to combine these ideas to IRL / adversarial imitation learning schemes. It is not clear that which expert demos we should use to show to the IRL algorithm. Should it be just one pleayer, or maybe k-most similar players to some seed player?

**Weaknesses:**

- Applications as of now are not very compelling. One option would be to take lichess.org data and ask some interesting research question about the player data that has not seen a sufficient answer. Then use this technique to answer it.
- Another option would be to integrate this technique to some IRL scheme.
- Authors cite speaker recognition in the related work, but did not study it more closely. In biometric recognition we have essentially similar task as in the present paper. Idea is to turn observed data (audio in speaker recognition) into fixed length vector and then by comparing these vectors obtain downstream recognition. Where the present paper goes wrong is that evaluation metric is identification accuracy, whereas in biometrics it is known that task is verification task and metric is ROC or DET curve and equal error rate (EER) summary statistic. Authors should use it, in addition to identification.
- Speaker verification also gives nice possibility for extensions to the current work. One idea back in the day was to do Baysian adaptation from the general speaker model to a specific speaker model. These ideas were also presented in the following paper https://arxiv.org/abs/2012.01244

**Questions:**

-

**Limitations:**

-

---

> ### Author Rebuttal · Authors · 2024-08-07
>
> Thank you for your insightful comments and questions! We respond to each one below, and in some cases refer to the general responses above.
>
> **Applications of our work**
>
> Please see our general response on the applications of behavioral stylometry; this includes two concrete use cases based on the experiments in our paper (Figs. 6 and 7) which have not been sufficiently answered by prior work. We also provide some analysis of the player style space in Lichess (chess) and Rocket League in Figs. 3, 4, and 5. Regarding other research questions about the player population/data, we would be happy to answer them if the reviewer has one in mind.
>
> **Connection to IRL**
>
> The connection to IRL and adversarial imitation learning are interesting directions to explore in future work. The focus of this paper is on efficiently learning individual behaviors and playing styles and creating a methodology to analyze and manipulate these styles. Our method for analyzing a player’s style applies heuristic functions to their observed decisions (e.g., Stockfish metrics like king safety, material imbalance, etc. for chess). These functions could be the components of a reward function learned through IRL. However, as the reviewer noted, IRL is typically applied to a single policy’s demonstrations with the goal of learning a reward function that distinguishes it from all other policies. Instead, we aim to learn a unique representation of every player policy simultaneously, and our representations (unlike reward functions) can generate player behavior.
>
> **Connection to biometric recognition, evaluation metrics**
>
> Thank you for deepening the connection to biometric recognition; we will add a deeper discussion of this in the paper. We can interpret our stylometry results using metrics like ROC by using a threshold for cosine similarity when comparing style vectors. The results in Figs. 3 and 5 suggest that we can separate individuals from the rest of the population by a large margin. Additionally, our stylometry accuracy results in Table 1 evaluate players that are both in the universe (“seen”) and outside (“unseen”). To verify this, we computed an ROC curve based on our Rocket League stylometry results, adapted to our multi-class setting by treating any match other than the actual player as a false result. **Please see the ROC curve in the pdf attached to our general response**, which shows an AUC of 0.87.
>
> **Related work on speaker verification**
>
> Thank you for pointing us to this related work; we will add discussion of it in our paper. One limitation of the Bayesian approach is that it creates a separate model for each individual, similar to the individual fine-tuning approach we compare against in Fig. 2. The paper by Kanervisto et al. [C] argues for using the states visited by a policy, rather than its actions, to characterize the policy’s behavior, citing computational costs and a lack of expressive power when using actions. Our work uses both states and actions to provide a method that is both expressive and scalable. Game states are the input features and actions are the labels when training our MHR models.
>
> [C] Kanervisto et al. General Characterization of Agents by States they Visit. Deep Reinforcement Learning Workshop, NeurIPS 2021.

---

> > ### Comment · Reviewer_JjNr · 2024-08-13
> >
> > Thank you for your detailed response. I am pretty much satisfied with it. I am inclined to raise my score by one.

---

### Official Review · Reviewer_Ff76 · 2024-07-14

**Soundness:** 3
**Presentation:** 3
**Contribution:** 2
**Rating:** 5
**Confidence:** 3

**Summary:**

In this paper, the authors propose to solve the problem of behavior stylometry, which is to identify the style of a player’s policy in the game, by regarding it as a multi-task learning problem. Each player’s style is a distinct task. Previous methods are either not scalable or not generative, in that they cannot predict the moves of a player given a query set of games played by that player. The authors aim to design a model capable of generating the moves of different players in a scalable manner. To do that, the authors trained a set of Low Rank Adapters (LoRAs) over a base model. The base model is trained with behavior cloning on the whole dataset, and the LoRAs are trained on a specific set of training player dataset separately. Additionally, a routing matrix is trained with LoRAs to specify the distribution over the LoRAs for each player. In this case, the authors claim that the routing matrix is a compact representation of the players’ skills, and encourage the adapters to learn different latent skills while preserved the shared knowledge within the base model. The authors also claim that the routing matrix supports few-shot learning, and induces a series of applicable benefits on stylometry, such as interpreting and manipulating the style of a player.

**Strengths:**

1. The paper is well written and clearly conveyed.
2. The proposed method achieves convincing results.
3. Their dataset is partitioned and rotated properly so that they can analyze the inter-player and intra-player consistency of the method, which provides strong convincing for the paper.

**Weaknesses:**

1. The main concern I have for this paper is the novelty of the proposed method. The base model for Chess is from a previous work called Maia, and trained with existing techniques such as LoRA and Polytropon, either of which is novel to me.
2. The significance of the problem setting, behavior stylometry, is not clear. The authors only have a very short introduction of what it is and its usage in the first section. However, the benefit of modeling accurate behavior stylometry is still vague. For example,  we have very high performance models in chess, even outperforming human players, then why do we need to model the style of human players via behavior stylometry? To me, accurately predict the moves of a player is not the ultimate goal of a game, the ultimate goal is to win the game. How can behavior stylometry help in winning the game? This should be better explained with more introduction and related experiments.

**Questions:**

1. How can behavior stylometry enhance human players? Why it is important to predict the moves of different human players, especially in the case of chess where super human models already exist? Can we analyze the style of AI methods with your method?
2. Why do you choose chess and Rocket League to test your approach? Are there any other potential games or environments applicable for your method?

**Limitations:**

As stated in the Weakness part, the analyze on the significance of behavior stylometry is limited. The authors work on a sub-problem of the game, and didn’t connect it back to the game itself.

---

> ### Author Rebuttal · Authors · 2024-08-07
>
> Thank you for your insightful comments and questions! We respond to each one below, and in some cases refer to the general responses above.
>
> **Novelty and new capabilities**
>
> Please see our general response above.
>
> **Significance of behavioral stylometry and human-like agents**
>
> We will add more explanation on the importance of behavioral stylometry and the motivation behind our work. Chess has seen superhuman models—both heuristic and AI-based—since 2007, all aimed at finding optimal moves with the sole objective of winning the game. However, these models are difficult to interpret, provide limited instructional value (that is not level-appropriate), and are unenjoyable to play against. Instead, we believe there is great value and use in creating human-like models and understanding how humans play; this is supported by a growing body of recent work on human behavior modeling (which includes the Maia line of work). Our work is the first to do this at the individual level at scale, while supporting style generation and synthesis.
>
> Please see our general response above on applications of behavioral stylometry and examples of how they can enhance human players.
>
> **Analyzing the style of AI agents**
>
> The same methodology used for learning, analyzing, and steering the style of human players, can also be used for AI models or any other playing agent. All we require is access to sample games played by the agent, which for AI/heuristic models can be generated easily via simulation. As Table 1 and Fig. 8 show, we only require a few games (e.g., 50) to learn an accurate style vector for the player.
>
> **Rationale for Chess and Rocket League, other environments**
>
> We chose Chess and Rocket League because they have a large, public collection of human games that span a diversity of skill levels and playing styles. Fig. 4 shows the diversity in playing styles using five randomly sampled players in chess. Since one of our goals is to push the scalability of generative modeling of individual behavior, we sought domains with thousands to tens of thousands of unique players at a minimum. In general, our methodology can be applied to any game, domain, or environment where individual-level data exists. For example, we are currently applying our style methodology to race simulators (Trackmania, iRacing), poetry data, and music data.

---

### Author Rebuttal · Authors · 2024-08-07

We thank the reviewers for their insightful comments, suggestions, and ideas. We provide responses to some common questions below as well as individual responses after each review.

**Novelty and new capabilities [Ff76, ALah]**

We do not claim novelty over the base models and PEFT techniques used in this paper, though we note that both base models were either designed (Rocket League) or modified (Maia) by us and outperform the state-of-the-art behavior cloning models. Instead, we believe our novelty comes from our use of PEFT techniques from NLP in a fashion that facilitates the learning of “style vectors". Style vectors are a novel concept that enable new and powerful capabilities, such as the ability to analyze player styles, create new styles, and steer (change) existing styles, all in a human-interpretable way.

**Applications of behavioral stylometry [Ff76, JjNr]**

Characterizing individual human behavior and style has many potential applications: we can identify weaknesses in a human’s play and develop personalized training material, develop better AI partners or teammates [A], and create more realistic or enjoyable playing experiences [B]. For example, the Maia bots on Lichess are the most popular bots by orders of magnitude, having played over 3.5 million games with humans. In addition, our interactions with several game studios and racing companies have shown a strong need for human-like agents, whether for personalized training, online teammate matching, or predicting how game physics/feature changes will affect human players.

Our paper already includes examples of how behavioral stylometry can enhance human players, based on our methodology for analyzing and steering player styles using style vectors (Section 4). We will update the text to call these out as potential applications, and describe two below:

1. Given any player in our universe who wants to improve, we can find a stronger player (higher Elo) with a similar playing style by comparing their style vectors. We can then linearly interpolate the weaker player’s style vector towards the stronger player’s vector using simple arithmetic. As Fig. 6 shows, these interpolation points result in new (synthesized) players whose playing strength interpolates roughly linearly between the two players. The weaker player can then play practice games against the interpolated players, observe their behavior on specific positions, etc., to learn and improve. Importantly, unlike learning from superhuman AI models, the interpolated players represent gradual, level-appropriate, and style-aware intermediate steps that the weaker player can learn from.

2. We can also take an existing player’s style vector and steer their gameplay along a particular heuristic, such as increasing their awareness of king danger or use of the bishop pair. As Fig. 7 shows, this steering results in new (synthesized) players with the desired properties. The player can then practice against or observe the behavior of these steered versions of themselves.

[A]  Hamade et al. Designing Skill-Compatible AI: Methodologies and Frameworks in Chess. ICLR, 2024.
[B]  Lichess bots page. Lichess, 2024

---

### Decision · Program_Chairs · 2024-09-25

**Decision:**

Reject

**Comment:**

This paper considers the problem of behavior stylometry by regarding it as a multi-task learning problem. In particular, it seeks to develop a generative approach that is scalable, which is a major challenge in existing methods. It develops a method by training multiple LoRA adaptors over a common base model to represent different styles. And then these adaptors / vectors are combined in different ways for action generation in the given style of a player. And they can also be used for interpreting and manipulating in the latent style space. The reviewers agree that this is an interesting but intuitive method. On the other hand, there are also concerns raised by the reviewers about the paper, such as slight lack of novelty and the significance of the problem being unclear. A better way to make this paper stronger is to extend the approach to more applications domains (beyond just chess games) and how the model can be used beyond stylometry.